# Object-aware Cropping for Self-Supervised Learning

**Shlok Mishra**[1]**, Anshul Shah**[2]**, Ankan Bansal**[1]**, Janit Anjaria**[1]**,**
**Abhyuday Jagannatha**[3]**, Abhishek Sharma, David Jacobs**[1]**, Dilip Krishnan**[4]
[1]**University of Maryland, College Park,**
[2]**Johns Hopkins University,** [3]**University of Massachusetts Amherst**
[4]**Google Research**
`{shlokm,dwj}@umd.edu, dilipkay@google.com`

Reviewed on OpenReview: `https://openreview.net/pdf?id=WXgJN7A69g`

## Abstract

A core component of the recent success of self-supervised learning is cropping data augmentation, which selects sub-regions of an image to be used as positive views in the self-supervised loss. The underlying assumption is that randomly cropped and resized regions of a given image share information about the objects of interest, which is captured by the learned representation. This assumption is mostly satisfied in datasets such as ImageNet where there is a large, centered object, which is highly likely to be present in random crops of the full image. However, in other datasets such as OpenImages or COCO, which are more representative of real world uncurated data, there are typically multiple small objects in an image. In this work, we show that self-supervised learning based on the usual random cropping performs poorly on such datasets (measured by the difference from fully-supervised learning). Instead of using pairs of random crops, we propose to leverage an unsupervised object proposal technique; the first view is a crop obtained from this algorithm, and the second view is a dilated version of the first view. This encourages the self-supervised model to learn both object and scene level semantic representations. Using this approach, which we call *object-aware cropping*, results in significant improvements over random scene cropping on classification and object detection benchmarks. For example, for pre-training on OpenImages, our approach achieves an improvement of 8.8% mAP over random scene cropping (both methods using MoCo-v2). We also show significant improvements on COCO and PASCAL-VOC object detection and segmentation tasks over the state-of-the-art self-supervised learning approaches. Our approach is efficient, simple and general, and can be used in most existing contrastive and non-contrastive self-supervised learning frameworks.

## 1 Introduction

In recent works on self-supervised learning (SSL) of image representations, the most successful approaches have used data augmentation as a crucial tool (Chen et al., 2020a; He et al., 2019; Grill et al., 2020; Tian et al., 2019; Caron et al., 2020b). Given a randomly chosen image sample, augmentations of the image are generated using common image transformations such as cropping and resizing a smaller region of the image, color transformations (hue, saturation, contrast), rotations etc. (Chen et al., 2020a; Gidaris et al., 2018). Of these augmentations, the use of cropping is clearly the most powerful (see Chen et al. (2020a), Fig. 5). This makes intuitive sense: cropping followed by resizing forces the representation to focus on different parts of an object with varying aspect ratios. This makes the representation robust to such natural transformations as scale and occlusion. The implicit assumption in this scheme is that the object of interest (classification or detection target) occupies most of the image and is fairly centered in the image, so that random crops of the image usually result in (most of) the object still being present in the cropped image. Such an assumption holds for "iconic" datasets such as ImageNet Krizhevsky et al. (2012). Forcing the resulting representations

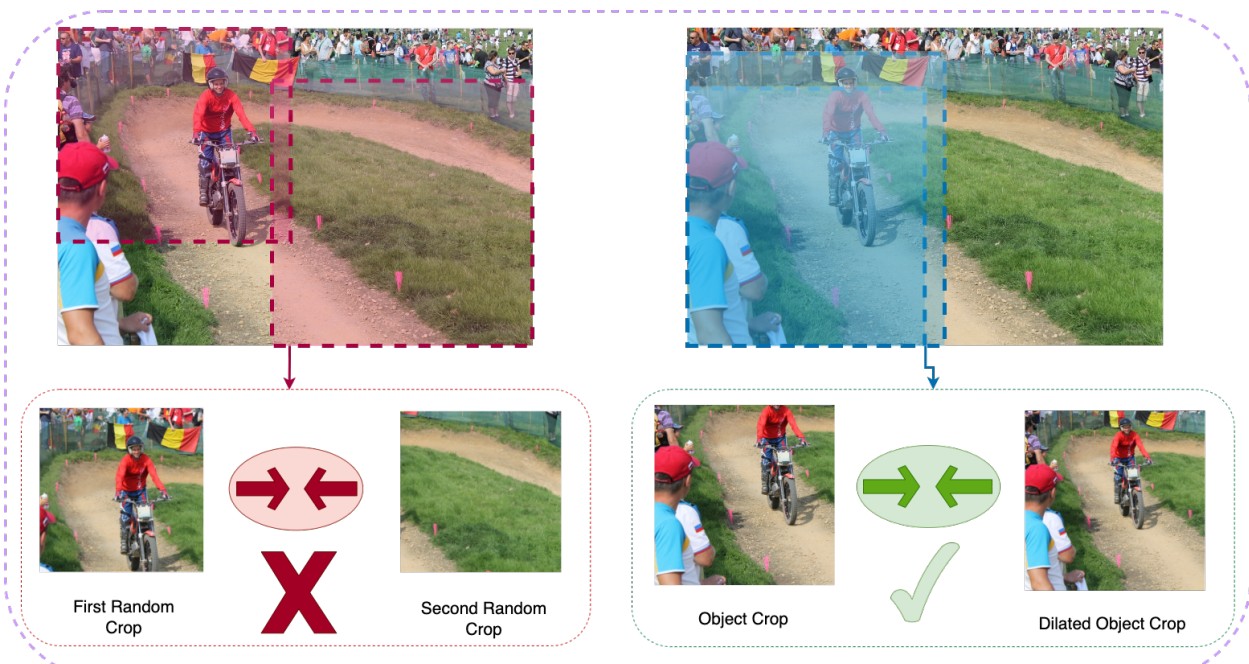

Figure 1: Illustration of object aware cropping. Top-Left: We show the original image with random crops overlaid. Bottom (red panel): Overlap between random crops tend to miss the object of interest. Top-Right: We show crops generated from the LOD Vo et al. (2021) algorithm and also the dilated object crop. Bottom-Right (green panel): We use LOD-based object-aware crops. These incorporates both object and scene information into the MoCo-v2 (or other SSL frameworks).

to be closer together maximizes the mutual information between the crops (also called *views*) van den Oord et al. (2018); Tian et al. (2019).

However, in the case of "non-iconic" datasets such as OpenImages Kuznetsova et al. (2020) and COCO Lin et al. (2014), the objects of interest are small relative to the image size and rarely centered, see Fig. 1. These datasets are more representative of real-world uncurated data. We find that the default random cropping approach (which we call *scene* cropping) leads to a significant reduction in performance for self-supervised contrastive learning approaches. For example, using the default pipeline of MoCo-v2 Chen et al. (2020b), we find that there is a gap of 16.5% mean average precision (mAP) compared to fully supervised learning. Other state of the art methods such as BYOL Grill et al. (2020), SwAV Caron et al. (2020b) and CMC Tian et al. (2019) perform poorly as well (see Table 2). As we show, the core problem here is that random scene crops do not contain enough information about (small) objects, causing degraded representation quality.

However, merely switching from scene-level crops to purely object-level crops does not exploit the correlations that exist between scenes and objects in most natural images. These correlations are helpful for downstream tasks Xiao et al. (2020). Keeping this in mind, we introduce ***O****bject-****A****ware ****C****ropping*(OAC), which applies a simple pre-processing step using the unsupervised object proposal method LOD Vo et al. (2021). LOD outputs multiple object proposal rectangles, one of which we pick at random as the first candidate region. We then expand (dilate) this rectangle to create a second candidate region. We finally employ random cropping within each of these rectangles to create a final pair of "positive" views that are used for the SSL loss. The use of random cropping reduces mutual information between the views, thereby making the pretext task harder for SSL losses, and improving final representation quality. We call this cropping approach "obj-obj+dilate" in the rest of the paper.

In addition to an SSL loss that uses multiple views, we introduce two additional unsupervised losses which leverage the LOD proposal introduced above. The first loss, which we call object localization, encourages the network's representations to carry information about objectness by predicting which patch features

| Method | Dataset | mAP |
|---|---|---|
| Supervised | - | 53.26 |
| InsDis(Wu et al. (2018b)) | ImageNet | 48.82 |
| MoCo( He et al. (2020) ) | ImageNet | 50.51 |
| PCL-v1(Li et al. (2021)) | ImageNet | 53.93 |
| PCL-v2(Li et al. (2021)) | ImageNet | 53.92 |
| MoCo-v2(Chen & He (2020)) | ImageNet | 50.67 |
| MoCHI(Kalantidis et al. (2020)) | ImageNet | 52.61 |
| MMCL(Shah et al. (2021)) | ImageNet | 50.73 |
| MoCo-v2 + OAC (Ours) | ImageNet | 53.96 |
| MoCo-v2 (Baseline) | OHMS | 54.74 |
| MoCo-v2 + OAC (Ours) | OHMS | **57.13** |

Table 1: We achieve superior performance on VOC detection when pre-training on our proposed **O**penImages **H**ard **M**ulti-object **S**ubset **OHMS** dataset as compared to ImageNet trained models by 6.52mAP. Our proposed OAC also helps ImageNet (2nd last row), but it helps much more on the proposed OHMS multi-object dataset. ImageNet baselines has been trained for 100 epochs Shah et al. (2021) and OpenImages model have been trained for same 100 ImageNet equivalent epochs.

contain the object (labels are determined by the extent of the LOD proposal). The second loss we add is the rotation prediction task where given a rotated object and dilated-object, we predict the rotation of the object. Rotation loss helps is learning better object level representations. Use of these losses leads to further improvements to downstream performance.

We conduct a number of experiments incorporating object-aware cropping, finding consistent improvements on state of the art self-supervised methods such as MoCo-v2 Chen et al. (2020b), BYOL Grill et al. (2020) and Dense-CL Wang et al. (2021) across varied datasets and tasks. We also propose **O**penImages **H**ard **M**ulti-object **S**ubset **OHMS**, which is a balanced subset of OpenImages and has images with atleast two different classes. We show by pre-training on the OHMS dataset using our OAC can give superior performance on object detection task (+6.1mAP, Table 1) as compared to pre-training on ImageNet which has been used extensively in the literature He et al. (2019); Chen et al. (2020a); Caron et al. (2020a).

## 2 Related Work

Recent progress in self-supervised learning, based on contrastive and non-contrastive approaches, has achieved excellent performance on various domains, datasets and tasks He et al. (2019); Chen et al. (2020b;a); van den Oord et al. (2018); Tian et al. (2019); Gidaris et al. (2020); Misra & van der Maaten (2019); Tian et al. (2020); Wu et al. (2018a); Grill et al. (2020); Gidaris et al. (2018); Larsson et al. (2017); Noroozi et al. (2018); Pathak et al. (2016). The top-performing methods have all used related ideas of pulling together "views" of a sample in representation space. Some of these approaches, in addition, use negative samples to add a "push" factor, and this is termed contrastive self-supervised learning. Theoretical and empirical studies have been published to better understand the behavior and limitations of these approaches Arora et al. (2019); Xiao et al. (2021); Purushwalkam & Gupta (2020); Tosh et al. (2021); Wang & Isola (2020); Yang et al. (2020); Chuang et al. (2020); Liu et al. (2021b); Kalantidis et al. (2020); Newell & Deng (2020); Cai et al. (2020).

A number of papers have observed that the default SSL approaches above (whether contrastive or not) perform poorly on uncurated datasets such as OpenImages Kuznetsova et al. (2020). To address this, recent works have used different workarounds such as knowledge distillation Tian et al. (2021), clustering Goyal et al. (2021), localization Selvaraju et al. (2020), unsupervised semantic segmentation masks Hénaff et al. (2021), pixel-level pretext tasks Xie et al. (2021), Instance localization Yang et al. (2021) and local contrastive learning Liu et al. (2021a). The common element among top-performing image-based SSL approaches, regardless of dataset, task or architecture, is their reliance on strong data augmentations such as random cropping, gaussian blurring, color jittering or rotations. These augmentations create meaningful positive views, and

other randomly sampled images in the dataset are used to create negative views in the case of contrastive SSL methods. SSL data augmentation pipelines are adapted from the supervised learning literature Cubuk et al. (2018); Zoph et al. (2020); Krizhevsky et al. (2012); Simard et al. (2003); DeVries & Taylor (2017); Cubuk et al. (2018); Zhang et al. (2018); Cubuk et al. (2019); Wu et al. (2019); Yun et al. (2019); Lim et al. (2019); Hataya et al. (2019). Chuang et al. (2022); Peng et al. (2022) also deal with issues in using random croping, but they don't report results by pre-training on multi-object datasets like COCO and OpenImages, which is our main focus in this paper. Secondly, their improvement on Detection Segmentation are only +0.3map while our improvement is +1.5 map by just changing the cropping strategy. Also, it's not very clear how to extend Chuang et al. (2022); Peng et al. (2022), since they rely upon bootstrapped models to generate better positives. And as we saw on OpenImages, bootstrapped models which use random cropping don't really perform well.

The closest work to our object cropping work is Selvaraju et al. (2020), which introduces a technique to choose crops around objects based on saliency maps Selvaraju et al. (2016), showing good improvements over the baseline of random crops for COCO datasets (see Table 5). As shown in our results, Obj-Obj+Dilate crop consistently performs better than Selvaraju et al. (2020) (Table 5). Our approach is also significantly simpler to incorporate into existing pipelines, requiring no change to the training, architecture or loss. Gansbeke et al. (2021) show that constrained multi-cropping improves performance of SSL methods: our approach can be incorporated into their pipeline to further improve performance.

# 3 Analysis of Self-Supervised Learning methods on the OpenImages Dataset

In this section, we first identify some limitations of state-of-the-art SSL methods such as MoCo-v2 He et al. (2019); Chen et al. (2020b), SwAV Caron et al. (2020b) and BYOL Grill et al. (2020) when pretraining on the OpenImages dataset. These methods have nearly closed the performance gap with supervised learning methods when pre-trained and linear probed on ImageNet Deng et al. (2009). However, the performance of these methods on OpenImages dataset (where images contain multiple small objects) has not been extensively studied. OpenImages Kuznetsova et al. (2020) encompasses images of complex scenes and several objects (containing, on average, 8 annotated objects per image). It consists of a total of 9.1 million images. To perform controlled experiments on the effectiveness of cropping on SSL method performance, we construct a subset of the OpenImages dataset called **OHMS O**penimages **H**ard **M**ulti-object **S**ubset. We construct the dataset as follows: We sample images that have labelled bounding boxes to enable comparisons with fully supervised learning. Secondly, we sample images with objects from at least 2 distinct classes to create a dataset that better reflects real-world uncurated data. Finally, we only consider class categories with at least 900 images to mitigate effects of imbalanced class distribution. After this processing, we have $212,753$ images present across 208 classes and approximately 12 objects per image on average.

We provide further details in the Appendix SecA.

## 3.1 Performance of SSL methods

We pretrain several SSL methods MoCo-v2, CMC Tian et al. (2019), SwAV Caron et al. (2020b) and BYOL Grill et al. (2020) on OHMS dataset. MoCo-v2, BYOL, and other recent state of the art SSL approaches all relying on *scene-scene* cropping of the same image to generate positive samples.

Table 2 shows our results. We see a significant difference in performance between fully supervised training and SSL approaches on the OHMS dataset, with a gap of 16.3 mAP points on average across the 4 SSL approaches considered. On ImageNet, the top-1 accuracy gap is considerably smaller with an average gap of only 8.5, nearly half that of OHMS. The last row shows the significant boost obtained by using our object-aware cropping approach, which we describe in the next section, with MOCO-v2. The gap between SSL and supervised training on OHMS is now the same as ImageNet.

| Model | OHMS (mAP) | ImageNet (Top-1 %) |
|---|---|---|
| Supervised Performance | 66.3 | 76.2 |
| CMC Tian et al. (2019) | 48.7 (-17.6) | 60.0 (-16.2) |
| BYOL Grill et al. (2020) | 50.2 (-16.1) | 70.7 (-5.5) |
| SwAV Caron et al. (2020b) | 51.3 (-15.0) | 72.7 (-3.5) |
| MoCo-v2 (Scene-Scene crop) | 49.8 (-16.5) | 67.5 (-8.7) |
| MoCo-v2 (Object-Object+Dilate crop) (Ours) | 58.6 (-7.7) | 68.0 (-8.2) |

Table 2: Classification results on OHMS and Imagenet. For each SSL method, we show in parentheses the gap to fully supervised training (same number of epochs). The last row shows that our proposed approach using obj-obj+dilate cropping reduces the gap on OHMS by nearly half compared to the baselines, improving over the scene-scene cropping based SSL methods by between 8.8 mAP points. We also observe improvements on ImageNet as well.

### 3.2 Analysis and Motivation

We conduct further experiments to better analyze the results seen in Table 2, and to motivate our proposed approach. Our experiments help to narrow down scene cropping as one main cause of the poor performance of SSL on OHMS, rather than other differences with ImageNet, such as object size, class distributions or image resolution.

**Object Size:** We compare MoCo-v2 performance to that of fully supervised learning, with both methods using scene-based cropping. Fig. 2 (left) shows that the performance gap between supervised learning and SSL methods does not vary significantly for objects of different sizes in OHMS. This suggests that once object sizes are below a threshold where scene cropping tends to ignore object information, MoCo-v2 performance is mostly independent of object scale.

**Long-tail Distribution:** Even after selecting at least 900 images per class, our OHMS subset has a significant variation in the number of images per class (from around 1000 to 60000). Fig. 2 (right) plots the performance of MoCo-v2 and supervised training as a function of the number of instances in each class. We do not see a significant change in relative performance as the number of instances in a class changes. This rules out long tails of the distribution as a cause for the poor absolute performance of MoCo-v2 on OHMS.

**Can ImageNet pre-training help?** We pre-trained a supervised model on ImageNet and then fine-tuned the final fully-connected layer on the OHMS dataset. We can see from the second column in Table 3 that this pre-training does not help to close the performance gap. One of the reasons that ImageNet pre-training does not help is that OHMS and ImageNet have significantly different class distributions (e.g. see Li et al. (2019) for a detailed analysis).

**Can resizing the images help?** We also experimented with resizing the images in OHMS to the same approximate size ($384 \times 384$) and aspect ratio as ImageNet. The result is shown in the third column of Table 3 confirming that controlling for image size does not help to close the gap.

**Can cropping on ground truth objects help?** We see from column 4 of Table 3 that using random cropping on ground truth object boxes does not help reduce the performance gap either. As we show later in Section4 that learning from both object and context is important for learning semantic information on multi-object datasets.

**Varying the lower scale of random resized crop:** MoCo-v2 (Chen et al. (2020b)) used scene crops whose size was chosen from a uniform distribution ranging from 20% to 100% of the ImageNet image size ($384 \times 384$). Since OHMS images are bigger and objects generally occupy a smaller fraction compared to ImageNet, we vary the lower bound for scene crops to measure the impact. The last six columns of Table 3 shows that varying the range of scene crop is insufficient to close the performance gap.

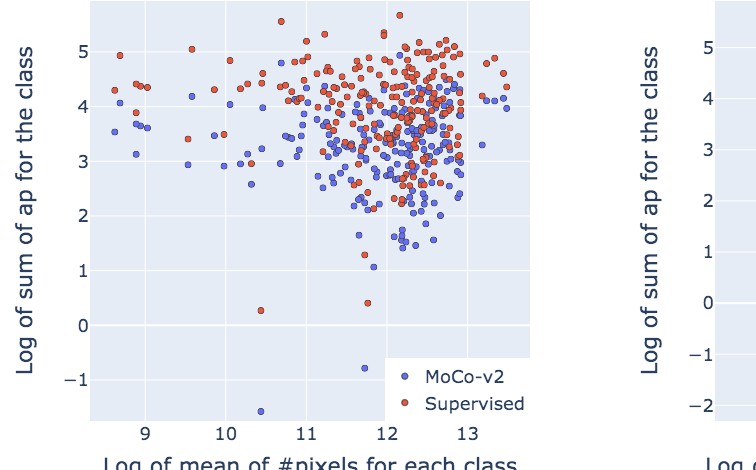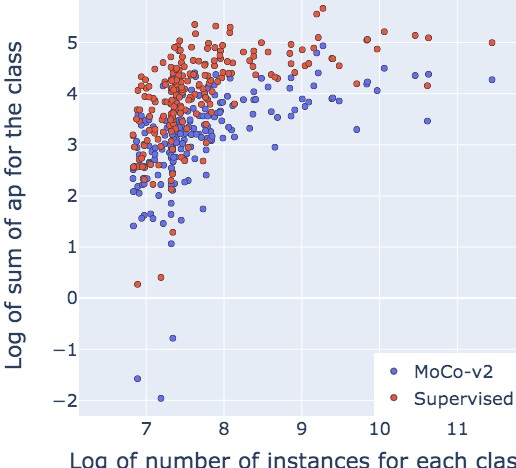

Figure 2: Analysis of OHMS data distribution. Left: Performance of supervised and MoCo-v2 pre-training as a function of the scale of the objects; we plot the log of the average of pixels against the sum of AP for each class. We see no discernible pattern of performance of MoCo-v2 or supervised learning as a function of object scale. Right: Performance of supervised learning and MoCo-v2 as a function of the number of instances in a class; we plot the log number of instances in a class against the sum of AP for that class. We do not see any discernible pattern of performance difference as a function of class size.

| | IN | Resize to | | Scene-Scene Crop | | | | | |
| Supervised | Pretraining | $(384 \times 384)$ | GT-Crop | 0.8-1.0 | 0.6-1.0 | 0.4-1.0 | 0.2-1.0 | 0.1-1.0 | 0.05-1.0 |
| --- | --- | --- | --- | --- | --- | --- | --- | --- | --- |
| 66.3 | 28.3 | 45.9 | 45.3 | 26.5 | 37.6 | 45.6 | 49.8 | 46.1 | 43.1 |

Table 3: Linear evaluation on our OHMS dataset with different pre-training strategies with MoCo-v2 (see Section 3 for details). Column 1 uses fully supervised learning on OHMS. We see that for self-supervised pre-training, no specific range of scene crops helps to close the large gap between SSL and supervised training. However, there is a sweet spot of scene crop range where MoCo-v2 performance is highest.

We conclude that the performance gap between supervised training and SSL training is likely due to the data augmentation, rather than characteristics of the image distribution. Further analysis experiments are provided in the appendix (Sec D).

## 4 Proposed Approach

In this section we first discuss how we add object-awareness to the loss, followed by two new loss functions which help in learning better object aware representations.

### 4.1 Object Proposals

To enable object-awareness, we consider an unsupervised object proposal model LOD Vo et al. (2021). LOD is a large scale unsupervised object proposal method Vo et al. (2021). The authors suggest a formulation of unsupervised object discovery as a ranking problem using distributed methods. In our experiments we use LOD to generate up to 10 proposals per image and select one object randomly among these proposals. The details of other faster semi supervised object proposals (which use object labels from VOC dataset for training) are present in appendix Section C. Since, our method is completely Self-Supervised we focus on using LOD as our object proposal generation method.

**Dilated object proposals (Obj-Obj+Dilate Crop):** Scene pixels spatially close to the object are more likely to have a positive correlation with the object. With this intuition, we generate the second view by dilating the the randomly selected LOD proposal. We dilate the box by 10% or 20% of the image size, followed by a random crop. Changing dilation gives us control over how much scene information is incorporated (a value of 10% works well in most cases). Note that the original and dilated boxes are both followed by a random crop, ensuring that the first view is not trivially included in the second view. The choice of which crop to use as query or key is arbitrary and either object or dilated object crops can be used as key and query.

**Different projection heads for Object and Dilated Object crops:** The projection head, introduced in Chen et al. (2020a) is an important component of most SSL methods. This is an MLP that maps representations from the encoder backbone to a lower-dimensional space where the loss function is applied. SimCLR Chen et al. (2020a) show that projection heads can remove information that may be useful for the downstream task, such as the color or orientation of objects. They show that by using this MLP, the last layer of network (i.e layer before mlp) can maintain more information. However Chen & He (2020) use a single projection head is used for both views, since both the views on ImageNet are object crops. We hypothesize that Object and Scene crops often contain different object orientation and color information; hence we propose to use different projection heads for scene and objects.

## 4.2 Loss Objectives

Following He et al. (2019), we use a momentum queue and optimize the model using the InfoNCE loss :

$$\mathcal{L}_{\text{moco}} = -\log \frac{\exp(q \cdot k^+/\tau)}{\exp(q \cdot k^+/\tau) + \sum_{k^-} \exp(q \cdot k^-/\tau)}. \tag{1}$$

Here $q$ is a query representation, $k^+$ is a representation of the positive (similar) key sample, and $\{k^-\}$ are representations of the negative (dissimilar) key samples. $\tau$ is the temperature hyper-parameter. We augment the standard contrastive loss with two additional losses to learn richer features for both objects and scenes. Next we describe both these losses in detail.

### 4.2.1 Rotation task $\mathcal{L}_{rot}$

In the standard rotation prediction pretext task Gidaris et al. (2018), an image crop is randomly rotated with an angle in set $\phi$. An MLP is then tasked at correctly predicting the rotation of a patch given its features extracted using the feature extractor $f$. Our object cropping strategy generates an object and a scene crop from a given image. We modify the standard rotation task to estimate rotation of the object crop wrt the scene crop. Specifically, we randomly rotate the object crop and extract features $z_{\text{obj}}$. The scene crop is kept as is and its features $z_{\text{scene}}$ are obtained by feeding the scene crop through feature extractor $f$. Note that rotation augmentations are applied to object crop in addition to the standard MoCo-style augmentations. The rotation prediction MLP takes as input the concatenated features and estimates the relative rotation. Our rotation loss is $\mathcal{L}_{\text{rot}}$ is a standard cross entropy loss with rotation labels as the targets. Different from the standard rotation prediction task which estimates absolute rotation, our approach estimates rotation of the object relative to the scene and complements the contrastive loss. Note that since we deal with in-the-wild datasets, estimating absolute rotation of a crop is ill-posed since a particular object might occur in a variety of poses thus leading to incorrect gradients.

### 4.2.2 Object localization task $\mathcal{L}_{loc}$

Since we are working with images coming from non-iconic datasets, the object could potentially occupy a small region of the image. We propose to add a pretext task of localizing an object inside the scene using features alone. Specifically in this task, given an image, we predict the spatial location of the object in the image. Here we take original image and divide the image into $p * p$ patches. We then take an object crop and mark all the patches where the object is present as 1 and other patches as 0 which we use as our label $Y$. Similar to rotation task we first extract features for the object proposals $z_{\text{obj}}$. For the original image we

| Model | Crops | Obj-Obj+Dilate | Scene-Scene | mAP |
|---|---|---|---|---|
| Supervised | - | - | - | 66.3 |
| MoCo-v2 | - | - | ✓ | 49.8 |
| BYOL | - | - | ✓ | 50.2 |
| MoCo-v2 | Ground Truth boxes | - | - | 58.9 |
| MoCo-v2 | Ground Truth boxes | ✓ | - | 60.2 |
| MoCo-v2 | LOD crops | ✓ | - | 59.1 |
| MoCo-v2 + Rotation + Localization | LOD crops | ✓ | - | red59.7 |
| BYOL | LOD crops | ✓ | - | 59.5 |

Table 4: Crop approaches on OHMS: using LOD crops to generate one view, and a dilated crop for the other positive, we are able to reduce the difference between SSL and Supervised Learning by close to 50% (compare the last two rows to the second row). Using ground-truth boxes to generate crops from OHMS improves the pre-training performance marginally compared to LOD crops. Our Obj-Obj+Dilate outperforms the Scene-Scene baseline by significant margin on the OHMS dataset.

apply all the same augmentations except random crop (since we want the full image and not the cropped version of the image). Then we obtain the features for the original image $z_{\text{ori}}$. We then pass the features through MLP layer. Finally the loss $\mathcal{L}_{ol}$ is two-class classification loss on concatenation of object features, original image features and labels $Y$.

**Why don't we see degenerate solution for object localization?** A natural question is why doesn't our model cheat and remember all the object location for all the object proposals. The answer to the question is the random cropping on the object proposals. Since we are using random crops on the object proposals, the label $Y$ will change every time according the random crop parameters. Hence the network cannot learn the object patch location and has to focus on the semantics of objects and scenes to figure out the spatial location of object in the scene. Exact implementation detail of the object localization task is shown in the appendix.

**Final loss function:** Our final loss function is

$$\mathcal{L}_{\text{oac}} = \mathcal{L}_{\text{moco}} + \mathcal{L}_{\text{rot}} + \mathcal{L}_{\text{ol}} \tag{2}$$

We simply combine all the losses and call it $\mathcal{L}_{\text{oac}}$ (**O**bject-**A**ware **C**ropping) and backpropagate on all of them.

## 5 Results

We created a subset of the OpenImages dataset with 212k images, as described in Section 3. We also experiment with the complete OpenImages (∼1.9 million images.) In addition, we perform pre-training on ImageNet Deng et al. (2009) and MS-COCO Lin et al. (2014). ImageNet (with 1.2M training images) has been extensively used and is the standard dataset used for benchmarking of SSL methods. MS-COCO has ∼ 118k training images and 896k labelled objects which is approximately 7 objects per image. For pre-training on MoCo-v2, we closely follow the standard protocol described in Chen et al. (2020b). We randomly select from 10 object proposals provided by LOD. All our training and evaluation is performed on a ResNet-50 He et al. (2015). For our baseline we use standard scene-scene crop, where we take two random crops in an image and treat them as positive views. This is the default approach used in MoCo-v2 and other SSL approaches. From our analysis in Section 3, this approach performs poorly on datasets such as OHMS.

As discussed, the Obj-Obj+Dilate uses a random crop on the object proposal or dilated version, to generate the final views. Since the object proposal itself is a small fraction of the image (e.g. in COCO, an object crop typically covers about 39% of the image), using the usual default lower value for the random crop range (usually 0.2) works poorly as it results in extremely small crops from the image. Therefore, we set the lower limit such that it matches the minimum sized crop in case of the usual scene crop ($s_{\text{min}} = \frac{0.2}{\text{average Object proposal size}}$) .

| Description | $AP$ | $AP_{50}$ | $AP_{75}$ | $AP_s$ | $AP_l$ | $AP_m$ |
|---|---|---|---|---|---|---|
| Supervised (Random Initialization) | 32.8 | 50.9 | 35.3 | 29.9 | 47.9 | 32.0 |
| Supervised (ImageNet Pre-trained) | 39.7 | 59.5 | 43.3 | 35.9 | 56.6 | 38.6 |
| MoCo-v2 Chen et al. (2020b) | 38.2 | 58.9 | 41.6 | 34.8 | 55.3 | 37.8 |
| BYOL Hénaff et al. (2021) | 38.8 | 58.5 | 42.2 | 35.0 | 55.9 | 38.1 |
| Dense-CL Wang et al. (2021) | 39.6 | 59.3 | 43.3 | 35.7 | 56.5 | 38.4 |
| CAST Selvaraju et al. (2020) (180K steps) | 39.4 | 60.0 | 42.8 | 35.8 | 57.1 | 37.6 |
| Self-EMD Liu et al. (2021a) (Uses BYOL) | 39.8 | 60.0 | 43.4 | - | - | - |
| MoCo-v2 + OAC (Ours) | **39.7** | **60.1** | **43.4** | **36.0** | **57.3** | **38.8** |
| Dense-CL + OAC (Ours) | **40.4** | **60.4** | **44.0** | **36.6** | **57.9** | **39.5** |
| MoCo-v2 + OAC (Using all losses)(Ours) | **40.7** | **60.9** | **43.9** | **36.9** | **58.3** | **39.6** |
| BYOL + OAC (Using all losses)(Ours) | **41.1** | **61.4** | **44.2** | **37.1** | **59.2** | **40.1** |
| Dense-CL + OAC (Using all losses)(Ours) | **41.4** | **61.5** | **44.7** | **37.5** | **59.5** | **40.4** |

Table 5: Object detection (first 3 columns) and Semantic Segmentation (last 3 columns) results on COCO dataset. All SSL models have been pre-trained on COCO dataset and then finetuned on COCO. All other methods are run for 90K, finetuning iterations. For any SSL method, we compare (BYOL, Moco-v2, Dense-CL) adding **O**bject-**A**ware-**C**ropping **(OAC)** cropping losses improves performance. We see further improvement by adding the proposed rotation and localization losses(last 3 rows).

| Model | $AP$ | $AP_{50}$ | $AP_{75}$ |
|---|---|---|---|
| Supervised (ImageNet-pretraining) | 56.8 | 83.2 | 63.7 |
| MoCo-v2 Scene-Scene crop (Chen et al., 2020b) | 51.5 | 79.4 | 56.4 |
| MoCo-v2 - Obj-Obj+Dilate crop ($\delta = 0.1$) (Ours) | **53.4** | **80.1** | **59.1** |
| MoCo-v2 - Obj-Obj+Dilate crop + Rotation ($\delta = 0.1$) (Ours) | **53.8** | **79.6** | **60.1** |
| MoCo-v2 - Obj-Obj+Dilate crop + Object Localization ($\delta = 0.1$) (Ours) | **54.1** | **80.6** | **60.2** |
| MoCo-v2 - Combined ($\delta = 0.1$) OAC (Ours) | **54.6** | **81.0** | **60.6** |

Table 6: Object detection results on VOC dataset (OHMS pre-training, using LOD proposals). All models have been pre-trained on OHMS and then fine-tuned on VOC. We can see that all of our proposed components improve upon the baseline scene scene crop and combining all of them improves upon the baseline by +3.1 mAP.

We evaluate the pre-trained models on classification (linear evaluation), object detection and semantic segmentation. For VOC object detection, COCO object detection and COCO semantic segmentation, we closely follow the common protocols listed in Detectron2 Wu et al. (2019). For VOC object detection, we evaluate on the Faster-RCNN(C4-backbone) Ren et al. (2015) detector on VOC `trainval07+12` dataset using the standard $1 \times$ standard protocol. For COCO-Object detection and semantic segmentation, we fine tune on the MaskRCNN detector (FPN-backbone) He et al. (2018) on COCO `train2017` split (118k images) with the standard $1 \times$ schedule, evaluating on the COCO 5k `val2017 split`. We compare to the state of the art SSL methods, including Self-EMD Liu et al. (2021a), DetCon Hénaff et al. (2021), BYOL Richemond et al. (2020), DenseCL Wang et al. (2021) and the default MoCo-v2 Chen et al. (2020b).

Table 2 and Table 4 shows results on OHMS dataset. We can see that Obj-Obj+Dilate crops outperform the baseline by 8.2 mAP, closing the gap between supervised learning and MoCo-v2 baseline by almost 50%. In Obj-Obj+Dilate crops, a dilated object crop would potentially contain the entire object and more scene information; therefore, the representation from the dilated object-crop contains complementary information from both the object and the scene. We also show in Table 4 an ablation with ground truth bounding boxes being used to guide the object cropping. This performs marginally better than the use of LOD crop, suggesting that a tight fit around the object is not necessary for improved representations.

| Description | $AP$ | $AP_{50}$ | $AP_{75}$ | $AP_s$ | $AP_l$ | $AP_m$ |
|---|---|---|---|---|---|---|
| COCO: MoCo-v2 (Scene-Scene crop) | 38.2 | 58.9 | 41.6 | 34.8 | 55.3 | 37.8 |
| COCO: MoCo-v2 + OAC (Ours) | **41.3** | **61.8** | **44.7** | **37.3** | **58.5** | **40.1** |
| VOC: MoCo-v2 (Scene-Scene crop) | 56.1 | 81.3 | 61.3 | - | - | - |
| VOC: MoCo-v2 + OAC (Ours) | **58.8** | **83.6** | **64.9** | - | - | - |

Table 7: Object detection (first 3 columns) and semantic segmentation (last 3 columns) results on COCO (first 2 rows) and VOC (last 2 rows). All SSL models have been pre-trained on complete OpenImages dataset(1.9 million images) for 75 epochs and then finetuned on COCO and VOC dataset.

| Model | Dataset | Crops | Obj-Obj+Dilate | Scene-Scene | mAP |
|---|---|---|---|---|---|
| Supervised | Full OpenImages | - | - | - | 74.0 |
| MoCo-v2 | Full OpenImages | - | - | ✓ | 50.5 |
| MoCo-v2 + OAC | Full OpenImages | ✓ | - | - | 62.1 |
| Supervised | Random Subset OpenImages | - | - | - | 74.0 |
| MoCo-v2 | Random Subset OpenImages | - | - | ✓ | 50.5 |
| MoCo-v2 + OAC | Random Subset OpenImages | ✓ | - | - | 62.1 |

Table 8: Results on full OpenImages and random subset of OpenImages. On full OpenImages dataset the difference between supervised learning and MoCo-v2 is still large, showing that random cropping is an issue not just on OHMS but also on the full dataset.

Table 5 shows results on object detection and semantic segmentation for COCO (by pre-training on COCO `trainval2017` datasets and finetuning on COCO). We train MoCo-v2, BYOL and Dense-CL models. Our MoCo-v2 Obj-Obj+Dilate cropping outperforms MoCo-v2 Scene-Scene baseline. Our proposed cropping is agnostic to the pre-training SSL method; we show results by adding our approach to Dense-CL Wang et al. (2021). We also outperform the CAST model Selvaraju et al. (2020) which also uses localized crops based on saliency maps: our approach is simpler and performs better by around 1.4 mAP. Table 1 (appendix) shows results of object detection on PASCAL-VOC. We pre-train on COCO and then fine-tune on VOC. OAC cropping outperforms the MoCo-v2 baseline by 3.2 mAP and the BYOL baseline by 2.5 mAP. Improved results on iconic datasets like Aircraft, Birds and Cars can be found in appendix (Table 4). Additional results on varying number of proposals used can be found in appendix (Table 6).

In addition to small OHMS dataset we also show results on full OpenImages dataset. Table 7 shows results on object detection and semantic segmentation for COCO and object-detection on VOC by pre-training on full OpenImages dataset Kuznetsova et al. (2020) (all 1.9 million images) for 75 epochs. We show improved performance over the baseline on both object detection and semantic segmentation tasks by using Obj-Obj+Dilate crops. Our proposed dilation method works not only for small multi-object datasets like COCO but also for datasets like OpenImages and performs well under a transfer learning setup.

**Results on ADE20K:** We also show results on ADE20k following Chen et al. (2017). The baseline MoCo-v2(COCO pre-training) gets 37.5 mIoU, while we get 39.2 mIoU. Similar to VOC and COCO, we see consistent performance improvement on ADE20k as well.

**Results on Full OpenImages:** We also report classification results on full OpenImages dataset i.e 1.7 million images and on 212k random subset in Table 8. For the full dataset pre-training has been done for 100 epochs and for the random subset the pre-training has been done for 200 epochs. We can see that on the full dataset, the gap between supervised learning baseline and MoCo-v2 is still large, while the gap on random subset is not as big.

**Results on ImageNet:** We also show improved results on ImageNet pre-training using object-aware cropping (Table 8 appendix) and MoCo-v2. For object detection on VOC2007, we see an improvement of 1.0

mAP; and a 0.5 mAP improvements for object detection on COCO. Our approach is thus adaptable to the pre-training dataset and SSL algorithm.

**Use of Multiple Projection Heads:** The use of different projection heads for each view on OpenImages classification gives us a boost of 1.1 mAP on Obj-Obj+Dilate crop. Pre-training on COCO and finetuning on VOC dataset for object-detection task gives a boost of 0.4 mAP. Hence using multiple projection heads results in a consistent improvement.

**Varying Dilation Parameter:** Table 3 (appendix) shows the effect of varying the dilation parameter. A sweet spot exists at a moderate dilation value of $\delta = 0.1$ for COCO object detection.

## 6 Conclusion

We have introduced object-aware cropping, a simple, fast and highly effective data augmentation alternative to random scene cropping. We conducted numerous experiments to show that object cropping significantly improves performance over scene cropping for self-supervised pre-training for classification, object detection and semantic segmentation on a number of datasets. The approach can be incorporated into most self-supervised learning pipelines in a seamless manner.

## 7 Acknowledgments

This work is supported[, in part,] by the US Defense Advanced Research Projects Agency (DARPA) Semantic Forensics (SemaFor) Program under [award / grant / contract number] HR001120C0124. Any opinions, findings, and conclusions or recommendations expressed in this material are those of the author and do not necessarily reflect the views of the DARPA. This research is also supported by the National Science Foundation under grant no. IIS-1910132.

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
