# OpenReview forum: "Object-aware Cropping for Self-Supervised Learning"
_TMLR — Accepted by TMLR_

### Review · Reviewer_Rh3t · 2022-10-01

**Summary Of Contributions:**

The manuscript presents an approach for contrastive self-supervised learning (SSL) targetting object detection tasks.

The main contribution is a novel way of selecting the positive views based on the LOD object discovery algorithm of Vo et al. by first selecting one of those proposals as the first positive view, and then creating a dilated version of that as  the second view. They complement this with two SSL losses, a secondary localization-oriented loss that leverages LOD scores and a rotation prediction loss.

They study how a number of common SSL algorithms perform when trained on a dataset that selected to be more complex than imagenet, and introduce OHMS, a subset of OpenImages that contains images with objects from at least two classes. It is a 200k image dataset with 208 classes and 12 objects per image. The


The authors show that when pretraining on OHMS with their method, they can get better results for downstream object detection datasets.

**Broader Impact Concerns:**

Added biases for this approach could be inherited via the proposals for the LOD algorithm.

**Requested Changes:**

R1) See weaknesses above: Results on ADE20K, another comparison in Tab2 and discussion on the computational cost of LOD.

R2) Table 1 is currently misleading should be split based on pretraining dataset. It is unfair to compare methods pretrained on Imagenet with ones trained on OHMS. At least one row for MoCo-v2 without OAC trained on the same OHMS dataset is needed as a fair baseline. This exists in other tables for other metrics, but should also be here for mAP on VOC to paint the whole picture

R3) The claims that Imagenet has a "a large, centered object" versus OpenImages being a dataset where "the objects of interest are small relative to the image size and rarely centered" are generally exargerated. There has been a number of works that show that ImageNet is not as iconic as generally though [Relabeling-Imagenet] while Figure 5 in [LVIS] shows that objects of interest in (the full) OpenImages dataset are indeed mostly centered. Of course the authors are right and OpenImages is more diverse than ImageNet, however unlike the section title,the authors do not use OpenImages for all the analysis, but they define a subset that is constructed to make the proposed algorithm shine. It would be great to tone down such claims and be explicit when the conclusions come from OHMS vs full OpenImages.
The two papers mentioned above are missing related works that should be cited and discussed at least in related work.

R4) It would be nice to report more results for the " Obj-Obj+Dilate crop" case without the added losses. Eg two more rows for Table 7, to see clearly what gain comes from that versus the added loss terms. I understand that the two are related to the LOD algorithm, but still it would be great if one can get gains by simply changing the cropping strategy and nothing else.

__Some clarifying questions__

Q1) What is the overlap between the classes of VOC and COCO and the 208 classes the OHMS dataset? How about Imagenet and COCO/VOC?

Q2) Sorry if I missed it but is there a train/val split for OHMS or are you performing evaluations and object detection on exactly the training images?

Q3) is the last row of Table 2 using the added losses (rotation and localization tasks)? If yes, what are the equivalent results without these added losses?

Q4) Can you clarify what you mean by "different projection heads for scene and objects" (end of Sec 3.1)  for MoCo-v2 which is generally the base SSL algorithm for the proposed approach? isnt the scene MLP (which i assume is the teacher) side also an EMA of the student side?

__References__

[LVIS] Gupta, A., Dollar, P. and Girshick, R., 2019. Lvis: A dataset for large vocabulary instance segmentation. In Proceedings of the IEEE/CVF conference on computer vision and pattern recognition (pp. 5356-5364).

[Relabeling-Imagenet] Yun, Sangdoo, Seong Joon Oh, Byeongho Heo, Dongyoon Han, Junsuk Choe, and Sanghyuk Chun. "Re-labeling imagenet: from single to multi-labels, from global to localized labels." In Proceedings of the IEEE/CVF Conference on Computer Vision and Pattern Recognition, pp. 2340-2350. 2021.


**Strengths And Weaknesses:**

__Strengths__

S1) the paper is very well written and structured and easy to read and understand

S2) OHMS is in general a very nice testbed for understanding performance of SSL algorithms in multii-object scenarios, and the analysis in Sec 2 clear and interesting.

S3) The method seems to give gains for different SSL methods, including Dense-CL which is a method targetting detection.

__Weaknesses__

W1) Results are limited to COCO and VOC. ADE20K would be a nice third dataset to test to.

W2) Although OHMS is in general a very nice testbed for understanding performance of SSL algorithms in multii-object scenarios, it is also kind of a "wish list" for the proposed approach, ie it contains only images that should be better represented with the proposed approach. Therefore Table 2 / Table 6 results should be taken with a pinch of salt. It would be really interesting to also compare performance of the proposed approach and other SSL methods when trained on a _random_ 200k-sized subset from all of Openimages in Table 2.

W3) Added computational cost of running LOD. How high is this cost? are the exact same representations used for this step? I assume this is a step that is run offline; the authors can report how much time it takes for each of the datasets they use (ImNet, COCO, VOC, OHMS, OpenImages)

---

> ### Author Response · Authors · 2022-11-09
> **Additional experiments and ablations.**
>
> **Results are limited to COCO and VOC. ADE20K would be a nice third dataset to test.** Here are the results on ADE20k. We follow the same protocol as [3].
> | Model |  mIoU |
> |-----------------|------|
> | MoCo-v2| 37.5  |
> | MoCo-v2 + OAC| 39.2 |
>
>
>  **It would be really interesting to also compare the performance of the proposed approach and other SSL methods when trained on a random 200k-sized subset from all of Openimages in Table 2.** Here is the comparison, when we used 212k random sized subset from OpenImages.
> | Model | mAP |
> |-----------------|------|
> | MoCo-v2 (Baseline)| 52.3  |
> | MoCo-v2 + OAC | 58.1  |
> | Supervised| 60.1  |
>
> **computational cost of running LOD**
> For OHMS, it took around 4.5 hrs to generate the proposals and for COCO it took us ~2 hours to generate the proposals.
>
> **At least one row for MoCo-v2 without OAC trained on the same OHMS dataset is needed as a fair baseline.** Thanks for the suggestion, we get 54.7 mAP by training scene-scene crop on OHMS dataset, we will add this result in the updated version.
>
> **Result on Full OpenImages**: As suggested we train MoCo-v2 on a full dataset i.e 1.7 million images for 100 epochs. For the testing set, we use the full test set of the OpenImages dataset. Here are the results after pre-training on full dataset.
> | Model | mAP |
> |-----------------|------|
> | MoCo-v2 (Baseline)| 50.5  |
> | MoCo-v2 + OAC| 62.1  |
> | Supervised| 74.0  |
>
>
>  **It would be nice to report more results for the " Obj-Obj+Dilate crop"**
>
> | COCO detection |  $\text{AP}$ |  $\text{AP}_{50}$ | $\text{AP}_{75}$ | COCO segmentation | $\text{AP}$ | $\text{AP}_{50}$ | $\text{AP}_{75}$ |
> |-----------------|------|------|------|-----------------|------|------|------|
> | MoCo-v2| 38.2    | 58.9| 41.6  | MoCo-v2   	|   34.8 | 55.3 | 37.8 |
> | MoCo-v2 + OAC  | 39.7 | 60.1 | 43.4 | MoCo-v2 + OAC | 36.0 | 57.3 | 38.8 |
> | MoCo-v2 + OAC (Using all losses)  | 40.7 | 60.9 | 43.9 | MoCo-v2 + OAC (Using all losses)  | 36.9 | 58.3 | 39.6 |
> | Dense-CL | 39.6    | 59.3| 43.3  | Dense-CL   	|   35.7 | 56.5 | 38.4 |
> | Dense-CL + OAC  | 40.4 | 44.0 | 60.4  | Dense-CL + OAC | 36.6 | 57.9 | 39.5 |
> | Dense-CL + OAC (Using all losses)  | 41.4 | 61.5 | 44.7 | Dense-CL + OAC (Using all losses)  | 37.5 | 59.5 | 40.4 |
>
> Below in the 1st row we show results by only using rotation and localization losses. We can see that using rotation and localization losses alone doesn’t work as well as when using all three losses.
> | Model |  mAP |
> |-----------------|------|
> | Using only rotation and localization loss| 44.3  |
> | MoCo-v2| 48.7  |
> | MoCo-v2 + OAC| 58.6  |
> | MoCo-v2 - OAC + Rotation + Localization| 59.7  |
>
>
> **Q1) What is the overlap between the classes of VOC and COCO and the 208 classes of the OHMS dataset? How about Imagenet and COCO/VOC?**: There is a 85% overlap between OpenImages and VOC, and between COCO and VOC the overlap is 95%. Between ImageNet and COCO, the overlap is 58%. Between ImageNet and VOC the overlap is 100%.
>
> **Q2) Sorry if I missed it but is there a train/val split for OHMS or are you performing evaluations and object detection on exactly the training images?**: To enable fair comparison with SOTA methods[1,2], the object detection was always done on the COCO dataset. For classification results on OHMS, we create a test set where we only sample images, if they have one of 208 classes which were selected while training.
>
> **Q3) is the last row of Table 2 using the added losses (rotation and localization tasks)? If yes, what are the equivalent results without these added losses?”**: No it was without adding additional losses. By adding additional losses the map improves to  59.7.
>
> **Q4) Can you clarify what you mean by "different projection heads for scene and objects" (end of Sec 3.1) for MoCo-v2 which is generally the base SSL algorithm for the proposed approach? isnt the scene MLP (which i assume is the teacher) side also an EMA of the student side?**
> Thanks for this question. A few clarifications : 1) As mentioned in Section 3, we randomly choose one of scene or object to pass through the query & key encoder. Thus, either the “scene” or “object” crop could be the “teacher”. 2) Different from the standard contrastive learning setup, we train two (query) MLPs , one each for object and scene crop and maintain EMA versions of the two on the student (key) side. We will make this clear in the final draft.
>
> [1] Dense-CL https://arxiv.org/abs/2011.09157
> [2] MoCo-v2 https://arxiv.org/abs/2003.04297
> [3] Rethinking Atrous Convolution for Semantic Image Segmentation https://arxiv.org/pdf/2203.11709

---

### Review · Reviewer_ANCp · 2022-10-07

**Summary Of Contributions:**

This work proposes a robust way to perform data augmentation, specifically, cropping, in contrastive learning. Along with the rotation loss and localization loss, the proposed loss function (OAC) achieves several empirical improvement across different benchmarks. Overall, the paper is well-written and the contribution is stated clearly.

**Broader Impact Concerns:**

The paper addresses a fundamental problem in self-supervised problem. I am not aware of any ethical issue in this work.



**Requested Changes:**

Overall, the paper is in a good shape. Having the following results would significantly strengthen the contribution.


1. Empirical experiments as stated in weakness sections. For instance, (1) Mocov2 + rotation + object localization (without LOD) (2) Empirical comparison with baselines such as [1] or [2] (3) Results on standard benchmarks such as ImageNet. By consolidating the experiments, I believe the contribution of the paper would be more convincing.

2. More discussion w.r.t. related works. As there are several similar works are addressing the same problem, it would be great to provide a paragraph to briefly go over the previous works.




**Strengths And Weaknesses:**

Strengths:

1. The overall improvement look good as Table 5 shows. It is also interesting to see BYOL fails to learn from noisy datasets. This is actually a very interesting observation that highlights the importance of negative samples. This direct is worth further explored in the future.

2. It is good to see dataset such as OHMS that mimics the real world distribution. The proposed dataset could potentially lead to several new contrastive learning methods.

Weaknesses:

1. The main motivation is that noisy cropping could degenerate the solutions of contrastive learning. It is unclear why one need rotation task and object localization task, as they seem to be orthogonal directions compared to contrastive learning. The only ablation study lie in Table 6, which adopts VOC dataset as the downstream task. It would be great to see how the proposed cropping approach performs without the additional two tasks on the COCO dataset.

2. The problem seems to be explored by several previous works that are not discussed here. For instance, [1] also proposes an object localization strategy to perform better cropping for contrastive learning. In [2], the same problem is addressed, where the authors propose a robust loss function to overcome the noisy cropping problem. Works with similar ideas can also be found in the other conferences. The existence of these previous works decrease the novelty of the proposed idea. In particular, the only baseline considered here is the Scene-Scene crop. More empirical comparisons to the previous works should be included to consolidate the experiments.

3. It is a bit unclear why we need two projection heads here. This makes it hard to tell how the proposed cropping method improves over the baselines. For instance, one can also augment MoCo-v2 with the rotation task and object localization loss to (potentially) improve the performance. Ideally, there should be more results that exclude all the additional losses and modifications, but simply changes vanilla cropping method to LOD.

4. In additional to OHMS, it would be great to see experiments with the standard evaluation setting, e.g., with pretraining on the ImageNet. It is tricky to evaluate the proposed approach on a self-constructed dataset, as there could be tunable factors during the process of dataset construction that makes the experiments

5. It is a bit odd to see the related works at the end of the paper. Similar to the second point, there are several related works that need to be discussed and compared. It makes sense to present an overview before the method section to give readers a glimpse about how similar problems are addressed.

6. In the appendix, what is the role of new p’(x) in (5), as p’ is not showed in the equation. If one use p’ to replace a part of p in (5), it is no longer a variational lower bound of the MI between X and C. Does this replacement create a new random variable? The theoretical analysis does not make the motivation clear.

Overall, the paper addresses an important problem in contrastive learning. The contribution could be potentially very useful to the field. Nevertheless, the ablation studies and comparison to previous works can be improved.

[1] Peng et al., Crafting Better Contrastive Views for Siamese Representation Learning, CVPR 2022

[2] Chuang et al., Robust Contrastive Learning against Noisy Views, CVPR 2022

---

> ### Author Response · Authors · 2022-11-09
> **Additional experiments and ablations.**
>
>
> **The problem seems to be explored by several previous works that are not discussed here** : Even though both [1] and [2] even though they deal with issues in using random crop, they don’t report results by pre-training on multi-object datasets like COCO and OpenImages, which is our main focus in this paper. Secondly, their improvement on detection & segmentation is only +0.3mAP while our improvement is +1.5 mAP by just changing the cropping strategy. Also, it’s not very clear how to extend [1] & [2], since they rely on using SSL model pre-trained on ImageNet to generate better positives. Hence these pre-trained models which work well on ImageNet may not extend well to OpenImages.
>
> **It is a bit unclear why we need two projection heads here. This makes it hard to tell how the proposed cropping method improves over the baselines. For instance, one can also augment MoCo-v2 with the rotation task and object localization loss to (potentially) improve the performance. Ideally, there should be more results that exclude all the additional losses and modifications, but simply changes vanilla cropping method to LOD.**
> Here is the ablation study when changing only the object crop and without the localization & rotation loss. We can see that by only changing the object crop the performance of the model improves by a fair margin ( +1.5map) and rotation and localization loss further improve the performance.
> | COCO detection |  $\text{AP}$ |  $\text{AP}_{50}$ | $\text{AP}_{75}$ | COCO segmentation | $\text{AP}$ | $\text{AP}_{50}$ | $\text{AP}_{75}$ |
> |-----------------|------|------|------|-----------------|------|------|------|
> | MoCo-v2| 38.2    | 58.9| 41.6  | MoCo-v2   	|   34.8 | 55.3 | 37.8 |
> | MoCo-v2 + OAC  | 39.7 | 60.1 | 43.4 | MoCo-v2 + OAC | 36.0 | 57.3 | 38.8 |
> | MoCo-v2 + OAC (Using all losses)  | 40.7 | 60.9 | 43.9 | MoCo-v2 + OAC (Using all losses)  | 36.9 | 58.3 | 39.6 |
> | Dense-CL | 39.6    | 59.3| 43.3  | Dense-CL   	|   35.7 | 56.5 | 38.4 |
> | Dense-CL + OAC  | 40.4 | 44.0 | 60.4  | Dense-CL + OAC | 36.6 | 57.9 | 39.5 |
> | Dense-CL + OAC (Using all losses)  | 41.4 | 61.5 | 44.7 | Dense-CL + OAC (Using all losses)  | 37.5 | 59.5 | 40.4 |
>
> Below in the 1st row we show results by only using rotation and localization losses. We can see that using rotation and localization losses alone doesn’t work as well as when using all three losses.
> | Model |  mAP |
> |-----------------|------|
> | Using only rotation and localization loss| 44.3  |
> | MoCo-v2| 48.7  |
> | MoCo-v2 + OAC| 58.6  |
> | MoCo-v2 - OAC + Rotation + Localization| 59.7  |
>
>
>
>
> **Pre-training on ImageNet**: Table 8 in supplementary shows results on ImageNet by using OAC. We can see that by just changing the crops to OAC, our performance improves by +0.8mAP on VOC. Due to  resource constraints, we have not added results by using Localization and rotation loss, which we will do in the subsequent versions.
> **It is a bit odd to see the related works at the end of the paper. Similar to the second point, there are several related works that need to be discussed and compared. It makes sense to present an overview before the method section to give readers a glimpse about how similar problems are addressed.**
> Thanks for the suggestion, we will move related work after introduction in the updated version.
>
> **In the appendix, what is the role of new p’(x) in (5), as p’ is not showed in the equation. If one use p’ to replace a part of p in (5), it is no longer a variational lower bound of the MI between X and C. Does this replacement create a new random variable? The theoretical analysis does not make the motivation clear.** P’(x) results from the new sampling scheme defined by our proposed cropping methods. So each of our cropping scheme can be viewed as enforcing a different probability distribution over x in (1). This is our interpretation, not an explicit formulation that derives the method.  We will update the theoretical analysis in the updated version.
>
> **Empirical experiments as stated in weakness sections. For instance, (1) Mocov2 + rotation + object localization (without LOD) (2) Empirical comparison with baselines such as [1] or [2] (3) Results on standard benchmarks such as ImageNet. By consolidating the experiments, I believe the contribution of the paper would be more convincing.**
> Comparison with [1],[2]: Both [1] and [2] even though they deal with issues in using random crop, they don’t report results by pre-training on multi-object datasets like COCO and OpenImages, which is our main focus in this paper.. Secondly, their improvement on Detection & Segmentation are only +0.3map while our improvement is +1.5 map by just changing the cropping strategy. Also, it’s not very clear how to extend [1] & [2], since they rely upon bootstrapped models to generate better positives. And as we saw on OpenImages, bootstrapped models which use random cropping don’t really perform well.

---

### Review · Reviewer_moVN · 2022-10-27

**Summary Of Contributions:**

The paper proposes a self-supervised representation learning method to learn from complex (multi-object) datasets, like OpenImages and COCO, which are more representative of real-world uncurated data. The proposed method optimizes three terms simultaneously: 1) a contrastive term, 2) a rotation prediction term, and a 3) localization term. Regarding the first term, the main technical novelty is the new image sampling procedure which, on one hand, focuses on objects in the scene (by leveraging unsupervised object proposals), and on the other hand, obtains the two views by using crops that are spatially close. The third loss term (localization) also seems to be novel, and seeks to train a model where object-centric crops (derived from the unsupervised proposals) can be localized in the full image. The proposed procedure resulted in significant improvements when compared to traditional methods like MoCo which learn from random crops. The paper shows convincing improvements on downstream tasks like detection and segmentations on COCO and VOC.

**Broader Impact Concerns:**

I have no ethical concerns regarding this paper.

**Requested Changes:**

Please, see the changes requested in the weaknesses section. Namely, the ablation study and the emphasis on the rotation and localization terms.

**Strengths And Weaknesses:**

Overall the paper is well written, addresses an important problem in self-supervised representation learning (that is learning representations from less curated and more naturally occurring data distributions), and achieves strong results. The main strengths were further described in the section above ("summary of contributions").

In my view, the main weakness of the paper is the following. The method uses 3 different self-supervised objectives. However, throughout the paper, the authors keep referring to only a single term (the contrastive between the object-aware crops) as the main contributor to the improved performance when training from complex images (eg on OpenImages). However, the authors do not report the results of MoCo/BYOL with the additional 2 loss terms. The only related result to this is in Table 6, which shows that adding the new crops, rotation, and localization (in this order) improves results on VOC, but there are no results showing the performance of a standard contrastive method with the additional rotation and localization pretext tasks (while foregoing the new crops). Thus, the claim that object-aware cropping is responsible for improved performance is still unsubstantiated. Even if the results of the above ablation show that the new crops lead to bigger improvements than rotation and localization by themselves, clearly, the other terms also play an important role in achieving state-of-the-art performance. So, I would still argue that the overall emphasis given to these components is lacking.

---

> ### Author Response · Authors · 2022-11-09
> **Additional experiments and ablations.**
>
>
> **report the results of MoCoL with the additional 2 loss terms without OAC.**
> We show ablation studies where we change only the object crop, without the localization & rotation losses. We add another ablation study by removing the OAC loss and using only rotation and localization losses.
> We can see that by only adding the object crop the performance of the model improves by a fair margin ( +1.5 mAP). The rotation and localization loss further improve the performance.
> | COCO detection |  $\text{AP}$ |  $\text{AP}_{50}$ | $\text{AP}_{75}$ | COCO segmentation | $\text{AP}$ | $\text{AP}_{50}$ | $\text{AP}_{75}$ |
> |-----------------|------|------|------|-----------------|------|------|------|
> | MoCo-v2| 38.2    | 58.9| 41.6  | MoCo-v2   	|   34.8 | 55.3 | 37.8 |
> | MoCo-v2 + OAC  | 39.7 | 60.1 | 43.4 | MoCo-v2 + OAC | 36.0 | 57.3 | 38.8 |
> | MoCo-v2 + OAC (Using all losses)  | 40.7 | 60.9 | 43.9 | MoCo-v2 + OAC (Using all losses)  | 36.9 | 58.3 | 39.6 |
> | Dense-CL | 39.6    | 59.3| 43.3  | Dense-CL   	|   35.7 | 56.5 | 38.4 |
> | Dense-CL + OAC  | 40.4 | 44.0 | 60.4  | Dense-CL + OAC | 36.6 | 57.9 | 39.5 |
> | Dense-CL + OAC (Using all losses)  | 41.4 | 61.5 | 44.7 | Dense-CL + OAC (Using all losses)  | 37.5 | 59.5 | 40.4 |
>
> Below in the 1st row we show results by only using rotation and localization losses. We can see that using rotation and localization losses alone doesn’t work as well as when using all three losses.
> | Model |  mAP |
> |-----------------|------|
> | Using only rotation and localization loss| 44.3  |
> | MoCo-v2| 48.7  |
> | MoCo-v2 + OAC| 58.6  |
> | MoCo-v2 + OAC + Rotation + Localization| 59.7  |

---

### Decision · Action_Editors · 2022-12-04

**Recommendation:** Accept as is

**Comment:**

All three reviewers recommend acceptance and I have no objection. The idea is clearly described and experiments are convincing. There were some initial concerns by the reviewers, and the revision addressed them adequately. I looked carefully through the reviews and the authors' responses, and have no remaining concern. Recommend accept as is.

**Audience:**

The topic is of great interest to the audience of TMLR

**Claims And Evidence:**

The reviewers agree that the claimed efficacy of object-aware cropping in SSL is well supported by convincing empirical results.